# Evaluation of an infection control protocol to limit COVID-19 at residential summer camps in 2021

**Tirzah Weiss**[☉], **Tate Reuter**[☉], **Evan Dowell**[☉], **Mitchell Singstock**[*☉], **Katherine Smith**[☉], **Jeffrey Schlaudecker**[☉]

Department of Community and Family Medicine, University of Cincinnati College of Medicine, Cincinnati, Ohio, United States of America

☉ These authors contributed equally to this work.
* singstmd@mail.uc.edu

**Data Availability Statement:** All supporting statistical files are available from the Dryad database (accession number doi:10.5061/dryad. b5mkkwhjr).

## Abstract

### Aim

To assess the effectiveness of an infection control protocol developed to mitigate the spread of COVID-19 at two multi-week residential summer camps in 2021.

### Subject and methods

Data were collected from 595 camp attendees and staff members at two wilderness camps in Northern Minnesota. Testing was undertaken in all unvaccinated campers before arrival at camp, on day 4 of camp, and in the event of respiratory symptoms. Campers were limited to cohorts during the first 4 days of camp and wore masks indoors. The number of positive COVID-19 cases measured the efficacy of the protocol.

### Results

The testing and cohorting protocol successfully prevented the spread of COVID-19 among campers and staff. During the first summer session, there were zero positive cases of COVID-19 among 257 campers and 127 staff. During the second summer session, compliance with the protocol limited the spread of COVID-19 to just three individuals of 266 campers and 129 staff. Maintaining cohorts at arrival limited spread from a single positive case to only two tent companions.

### Conclusion

The testing and cohorting protocol limited the spread of COVID-19 among residential summer wilderness campers and staff. Post-arrival testing ensured newly acquired virus was limited in spread before COVID-19 precautions were relaxed on camp day 5. A strict evidence-based cohorting protocol limited in-camp spread and allowed for a successful summer camp season. The usefulness of this protocol with an evolving pandemic, increasing vaccination rates, and virus variants could have implications for future practice.

**Funding:** The authors received no specific funding for this work.

**Competing interests:** The authors have declared that no competing interests exist.

**Abbreviations:** ACA, American Camp Association; CDC, Centers for Disease Control and Prevention; COVID-19, Coronavirus Disease 2019; PCR, Polymerase Chain Reaction; uC-URI, non-COVID Upper Respiratory Infection.

## Introduction

Summer camps are important for youth development: being outside and unplugging is critical perhaps now more than ever. Research conducted by the American Camp Association (ACA) has shown that summer camps produce statistically significant increases in camper self-esteem, independence, and leadership capacity [1]. In the summer of 2020, young people suffered a great loss as many camps could not provide the typical experience to campers. Indeed, 58% of overnight camps, 29% of day camps, and 58% of combo day/overnight/rental camps that were included in a survey distributed by the ACA did not offer in-person camp programming [2].

Opening a summer camp for participants comes with the risk of COVID-19 infection and transmission. However, children typically experience less severe COVID-19 symptoms [3] and are less likely to be index patients for an outbreak [4]. Children are also more likely to be asymptomatic, mild, or moderate in their disease severity, making underdiagnosis a possibility [3]. Mitigation of spread is critical to protect children but also to protect the older and vulnerable populations with whom they live and interact.

Published experiences of camps that operated in 2020 experienced varying degrees of COVID-19 transmission [5–7]. Four overnight summer camps in Maine limited COVID-19 activity through pre-arrival quarantine, testing, and symptom screening before and after arrival, camper cohorting, masking, social distancing, increased hygiene, and enhanced cleaning and disinfecting practices. Outdoor activity was prioritized, and campers were monitored daily for symptoms associated with COVID-19. Camp staff were also unable to leave campus. Despite three positive tests after camp arrival (at three different camps), these comprehensive measures were successful in preventing outbreaks [5].

Less successful was a summer camp in Georgia, which closed just six days after opening to campers, reporting 260 confirmed positive cases [6]. While this camp heeded many of the Centers for Disease Control's (CDC) recommendations, cloth masks were not required amongst campers, and doors and windows were not opened to increase ventilation. Additionally, campers at this camp regularly engaged in intense cheering and singing, a known transmission route of COVID-19.

The vulnerability of young populations to COVID-19 transmission is also related to vaccination status. Vaccination is among the most successful methods to prevent symptoms and spread of COVID-19 by bolstering herd immunity [7]. While healthy children are generally less likely to suffer severe symptoms of COVID-19 when compared to those with pre-existing medical conditions and the elderly, they are still vulnerable to potentially serious complications of illness and can spread the coronavirus throughout their communities. In children in the United States, the baseline rate of hospitalizations from COVID-19 was 8 in 100,000; however, these rates increase in children with obesity, chronic lung disease, and/or who are black [8]. During the summer of 2021, the delta (B.1.617.2) variant represented between 50% and 90% of all new cases [9]. The predominant mRNA vaccines were found to be less effective at preventing transmission from the delta variant compared to the previously predominant alpha variant (96% vs. 87%) [10]. However, vaccinations can only be effective if administered to the general population. Two major limitations for broader vaccination during the Summer of 2021 were government approval and vaccine hesitancy. The FDA (Food and Drug Administration) approved the Pfizer-BioNTech vaccine for 11–15-year-olds in May 2021 just before the start of our summer camp. Vaccines were approved for 5–10-year-olds in October 2021 [11]. During June 2021, 33.4% of surveyed Americans were "hesitant" towards vaccination, which is greater than the global average of 24.8% and increased from June 2020 when 24.6% of Americans were "hesitant" [12]. At the time of this study, approximately 12% of the population in

**Table 1. April 2021 CDC-recommended procedures for camps with incomplete camper vaccination.**

| CDC-Recommended Procedures, Spring 2021 | |
|---|---|
| • Promoting vaccination<br>• Wearing well-fitting masks<br>• Physical distancing<br>• Cohorting of campers<br>• Increased handwashing<br>• Covering coughs and sneezes<br>• Increased cleaning practices<br>• Remaining at home if sick/symptomatic<br>• Testing if symptomatic or unvaccinated close contact<br>• Contact tracing<br>• Screening testing | • Avoidance of crowded/poorly ventilated areas<br>• Unvaccinated camper masking during outdoor events when physical distancing not possible<br>• Universal masking to aid adherence to prevention strategies<br>• Cohorts for overnight camps should be those campers sleeping in the same cabin<br>• Following interim public health recommendations for vaccinated individuals |

Minnesota was "hesitant" and 69% was fully vaccinated [13]. After being closed in 2020 due to COVID-19, two summer camps in Minnesota re-opened in 2021. COVID-19 protocols for these Minnesota summer camps were developed in April and May of 2021 based on published recommendations. In a camp population with incomplete vaccination, multiple infection-mitigation policies and procedures were recommended (see Table 1) [14]. The purpose of this study was to assess the effectiveness of an infection control protocol developed to mitigate the spread of COVID-19 at two multi-week residential summer camps in 2021. The primary outcome of interest, therefore, is COVID-19 transmission rates within the summer camp.

## Methods

A medical advisory team was created to develop a strategy for preventing a COVID-19 outbreak at two residential wilderness summer camps, one all-male and the other all-female. Each camp had two sessions, each 4 weeks long, with a 2-day break between sessions. During each session, campers would embark on a 3 to 21-day wilderness trip in the northern United States.

The advisory team, made up of volunteer medical professionals with camp affiliations, including one of the authors (JS), developed a list of protocols based on research and recommendations by the CDC, ACA, National Institute of Health, and Minnesota Department of Health. The new COVID-19 policies were shared between the two camps. These policies were communicated to campers and their families during Spring 2021. After reading and understanding this information, families were required to sign a COVID-19 waiver indicating that they would abide by the new policies. This study was evaluated by the University of Cincinnati Institutional Review Board and deemed exempt as not human subject research.

### Vaccination policy

All staff members were required to be fully vaccinated against COVID-19 with two doses of either the Pfizer-BioNTech or Moderna vaccine prior to traveling to camp. Campers were not required to be vaccinated against COVID-19 but were highly encouraged to receive the vaccine [10]. Due to the age range of campers, it was not possible for all campers to be vaccinated.

### Quarantine and pre-camp COVID-19 testing policies

All campers and their families were requested to wear a face mask and avoid public gatherings during the two weeks before arriving at camp. Five days prior to traveling to camp, parents or guardians were required to administer an at-home COVID-19 PCR saliva test for their camper. Training for saliva collection was provided with the test package instructions. Samples were then mailed to Hennepin Healthcare in Minneapolis, Minnesota utilizing prepaid

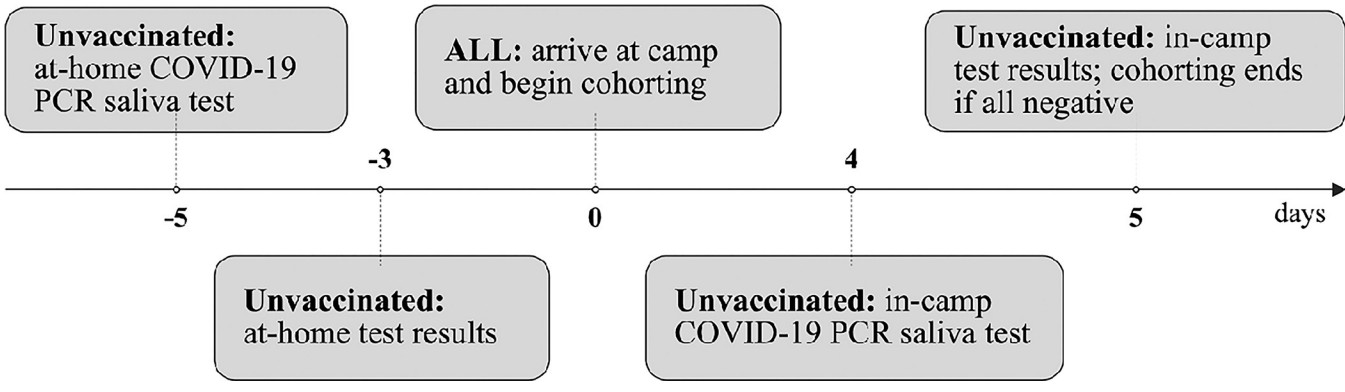

**Fig 1. COVID-19 PCR testing timeline for unvaccinated campers.**

shipping envelopes. Results were electronically communicated to parents 24–48 hours later via email. Those campers who submitted proof of full vaccination at the time of testing were not required to undergo COVID-19 PCR screening.

The day prior to travel, families were required to complete a COVID-19 health check online. Families were also provided with a KN95 mask that their camper was required to wear while traveling to and from camp. A complete timeline of camper testing requirements is outlined in Fig 1.

## In-camp cohorting

Upon arrival at camp, all campers were screened by medical staff for COVID-19 symptoms. Any camper that exhibited symptoms of COVID-19 received a rapid antibody test via nasal swab for COVID-19 and were quarantined. If the test was positive, a family member was required to pick up the camper by car.

For the first four days of camp, campers remained in cohorts based on the cabins in which they slept. During the primary quarantine period, campers were required to wear masks when interacting with others outside their cohort. While eating, showering, or brushing teeth, proper social distancing was maintained. Dining halls were limited to 50% capacity and campers were fed in two shifts to accommodate social distancing. Staff members were taught the in-camping cohorting guidelines prior to each camp session. Reminders were provided during each meal. Additionally, staff members were encouraged to correct campers and staff members who were not following masking or cohorting guidelines.

Although fully vaccinated, all staff members also wore a mask when interacting with campers outside their cohort or when inside public spaces. Throughout the summer, staff members were required to wear a mask when leaving camp property and were not allowed to visit restaurants or other high-traffic environments such as movie theaters. Although staff members at both camps interacted with each other, there was no interaction between campers at the girls and boys camps.

## In-camp COVID-19 testing

On day four of both camp sessions, all participants who were not fully vaccinated were administered a second COVID-19 PCR test via saliva collection. All saliva collection was overseen by the medical staff or by staff members who were directly trained by the medical staff on how to collect saliva for PCR testing. The tests were driven to Hennepin Healthcare in Minneapolis, Minnesota to expedite results. Once all tests came back negative, campers were free to move

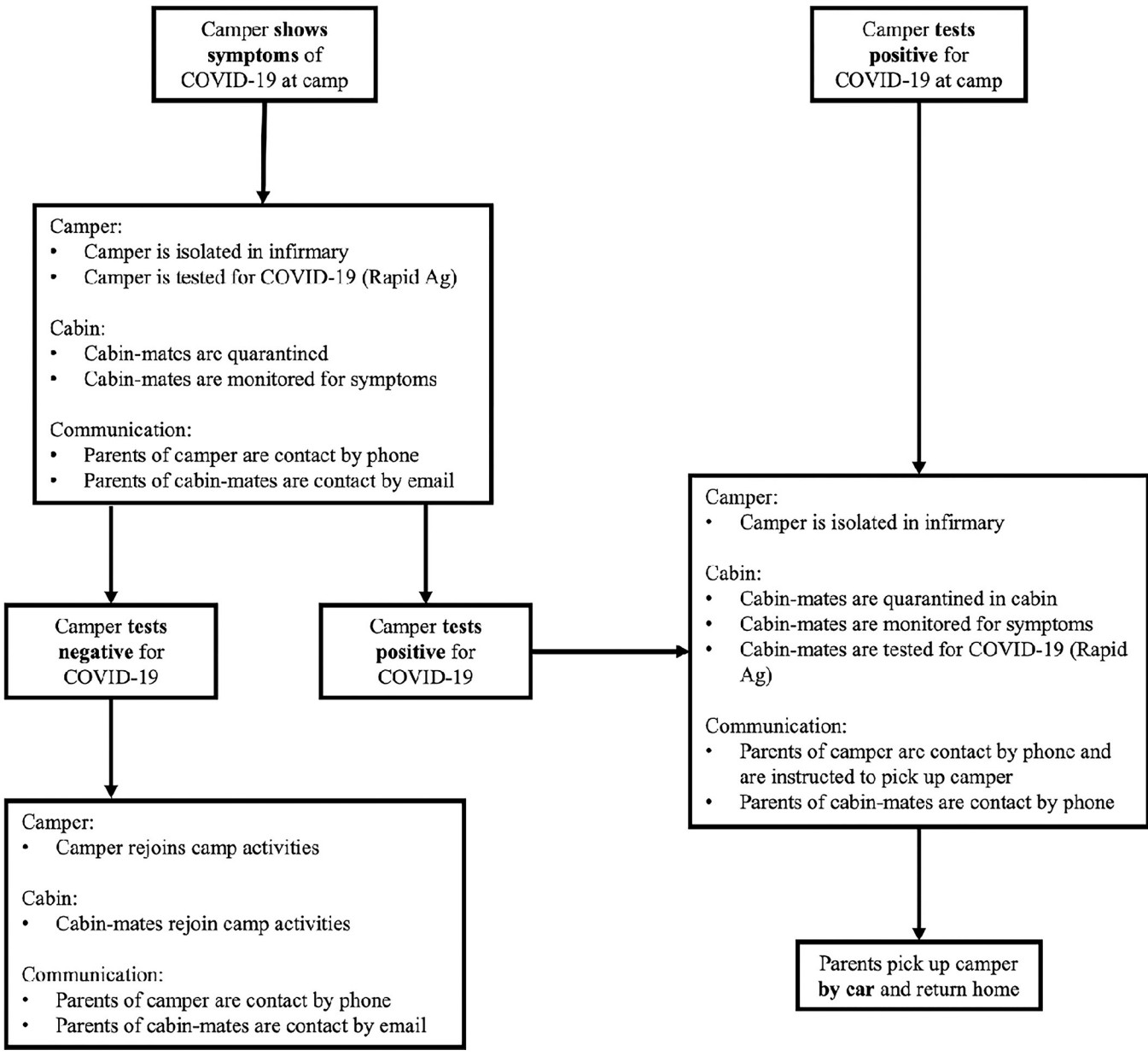

**Fig 2. A decision-making policy if a camper showed COVID-19 symptoms while at camp.**

outside of their cohorts and masks were no longer required. Campers and staff were encouraged to continue to screen themselves for COVID-19 symptoms and to report to the infirmary if they felt ill.

If a camper showed symptoms of COVID-19 at camp, the camper was immediately isolated and tested for COVID-19 via rapid antigen test. Cabin-mates were then isolated and monitored for symptoms. Any camper that tested positive would be sent home (Fig 2).

Each wilderness trip was equipped with one extra tent, a satellite phone, a geo-locator, and two COVID-19 rapid tests to safely deal with a symptomatic camper. If a camper showed symptoms of COVID-19 while on a wilderness trip, they would be isolated in the extra tent, and the medical advisor contacted via the satellite phone. If appropriate, a rapid COVID-19

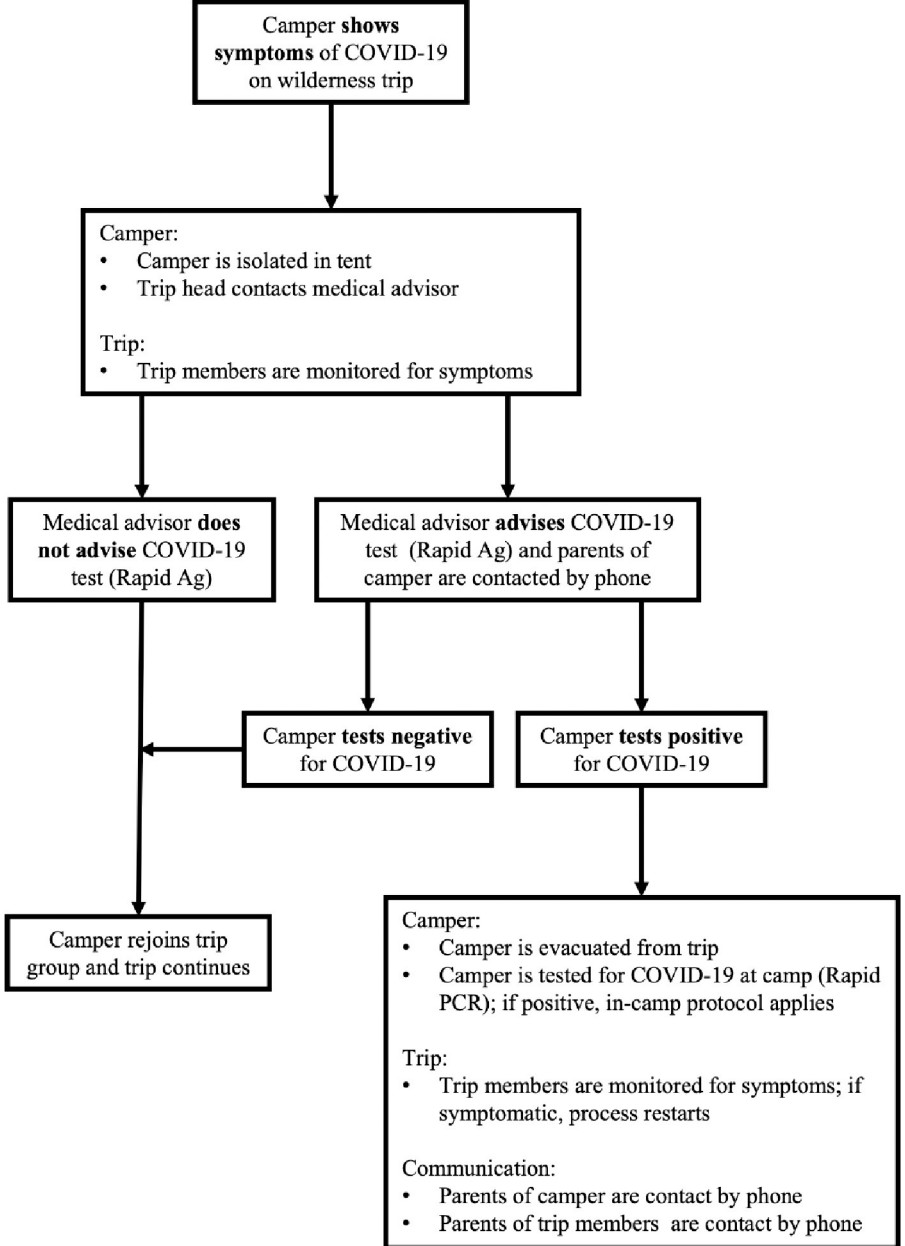

**Fig 3. A decision-making policy if a camper showed COVID-19 symptoms while on a wilderness trip.**

test would be administered. Any camper that tested positive while on a wilderness trip would be evacuated from the trip and sent home. The geo-locator aided base camp staff to locate the wilderness trip and coordinate a pickup point (Fig 3).

## COVID-19 testing and registry

A deidentified registry of all planned in-camp testing and symptomatic testing of staff and campers was maintained for each camp separately. Positive test results were communicated to the Minnesota Department of Health.

**Table 2. Characteristics of campers and staff at the residential summer camps.**

| | Boys Camp | Girls Camp |
|---|---|---|
| Staff | 75 | 54 |
| Campers | 238 | 228 |
| First Session | 141 | 118 |
| Second Session | 147 | 122 |
| Intersession | 50 | 12 |
| **Camper Ages** | | |
| <8 | 0 | 2 |
| 8–10 | 14 | 14 |
| 10–12 | 47 | 58 |
| 12–14 | 89 | 68 |
| 14–16 | 51 | 54 |
| 16–18 | 35 | 25 |
| >18 | 2 | 7 |
| **Vaccinated** | | |
| Yes | 114 | 87 |
| No | 124 | 141 |
| **Transportation** | | |
| Plane | 105 | 102 |
| Car | 80 | 78 |
| Shuttle | 49 | 46 |
| Unrecorded | 4 | 2 |
| **Home Region** | | |
| Northeast | 15 | 21 |
| South | 40 | 20 |
| Midwest | 129 | 148 |
| West | 52 | 36 |
| International | 2 | 3 |

## Results

### Camper characteristics

A total of 466 campers attended the two summer camps during June and July of 2021; 238 (51.1%) were male and 228 (48.9%) were female (Table 2). 50 (21.0%) male and 12 (5.3%) female "intersession" campers stayed for both sessions covering eight weeks. Over the course of the summer, 201 (43.1%) campers were fully vaccinated and 265 (56.9%) were unvaccinated. 207 (44.4%) of campers traveled to camp by plane. Additionally, most campers traveled to camp from the Midwest region (277, 59.4%). There was a total of 129 staff members between the two summer camps, 75 (58.1%) at the boys camp and 54 (41.9%) at the girls camp. A total of 5 campers were from countries outside the United States. No additional testing or quarantine was required of international travelers or those traveling by plane. A summary of camper characteristics is provided in Table 2.

### COVID-19 vaccination status

The FDA authorized the use of COVID-19 vaccinations in adolescents aged 12–18 in May 2021 (FDA 2021). As this authorization was close to the start of camp in mid-June, only 89 of 259 (34.4%) campers (89, 259 34.4%) were vaccinated for the first session. However, the camp

**Table 3. COVID-19 vaccination status of campers and staff at the residential summer camps.**

| | Boys Camp | | | |
| --- | --- | --- | --- | --- |
| | First Session | | Second Session | |
| | Vaccinated | Not Vaccinated | Vaccinated | Not Vaccinated |
| Staff | All (N = 75) | 0 | All (N = 75) | 0 |
| Campers 16+ | 26 | 8 | 25 | 7 |
| Campers 12–15 | 34 | 44 | 61 | 19 |
| Campers <12 | 0 | 29 | 2 | 33 |
| Camper Total | 60 | 81 | 88 | 59 |
| | Girls Camp | | | |
| | First Session | | Second Session | |
| | Vaccinated | Not Vaccinated | Vaccinated | Not Vaccinated |
| Staff | All (N = 52) | 0 | All (N = 54) | 0 |
| Campers 16+ | 16 | 8 | 9 | 2 |
| Campers 12–15 | 13 | 47 | 49 | 19 |
| Campers <12 | 0 | 34 | 2* | 41 |
| Camper Total | 29 | 89 | 60 | 62 |

directors encouraged families and campers to get vaccinated as soon as possible, and by the second session in July, 148 of 261 (56%) were fully vaccinated.

During the first session, 43.6% of boys aged 12–15 were vaccinated while 21.7% of girls aged 12–15 were vaccinated. However, during the second session, this proportion increased to 76.3% of boys and 72.1% of girls aged 12–15. The 16–18 age group had the highest proportion vaccinated. Of boys aged 16–18, 76.5% were vaccinated during the first session and 78.1% were vaccinated during the second session. Of girls aged 16–18, 66.7% were vaccinated during the first session and 81.8% were vaccinated during the second session. Of note, one female camper* under the age of 12 was vaccinated during session 1 because she was involved in a clinical trial, and another was vaccinated just prior to her 12th birthday (Table 3). "Intersession" campers who were present for both first and second sessions were counted for both session vaccination data.

## COVID-19 PCR testing

During the first session, 100% of campers tested negative for COVID-19 with a PCR test at home and on day 4 of camp. Again, during the second session, 100% of campers tested negative for COVID-19 with a PCR test at both timepoints (Table 4). There were several reasons for differences in the number of campers tested at each timepoint, including reaching full vaccination status at the time of in-camp testing. If a camper remained at camp between sessions, only in-camp testing was repeated. One test at the girl's camp was lost between administering the test and receiving the results†. She was presumed negative due to lack of symptoms and adherence to proper quarantining procedures prior to the day 4 in-camp testing timepoint.

Three campers were sent home before the in-camp testing timepoint, as described in the *Symptomatic Testing for COVID-19* section below. One camper at the girls camp had COVID-19 within 90 days of both timepoints and was exempt from testing. Of note, two rapid antigen tests were conducted with negative results at the boys camp during the second session in-camp testing due to inconclusive PCR results. A summary of COVID-19 PCR testing is found in Table 4.

**Table 4. COVID-19 PCR testing results for unvaccinated campers.**

| | Boys Camp | | | |
| --- | --- | --- | --- | --- |
| | First Session | | Second Session | |
| | At-Home | Camp Day 4 | At-Home | Camp Day 4 |
| | Pos/Neg | Pos/Neg | Pos/Neg | Pos/Neg |
| ≥12 yo | 0/81 | 0/81 | 0/23 | 0/18 |
| <12 yo | 0/32 | 0/32 | 0/33 | 0/33 |
| Total | 0/113 | 0/113 | 0/56 | 0/51 |
| | Girls Camp | | | |
| | First Session | | Second Session | |
| | At-Home | Camp Day 4 | At-Home | Camp Day 4 |
| | Pos/Neg | Pos/Neg | Pos/Neg | Pos/Neg |
| ≥12 yo | 0/74 | 0/62[†] | 0/24 | 0/17 |
| <12 yo | 0/34 | 0/34 | 0/39 | 0/41 |
| Total | 0/108 | 0/96 | 0/63 | 0/58 |

## Symptomatic testing for COVID-19

At the boys camp, 2 vaccinated staff members during the first session showed symptoms and were administered a rapid test. Both staff members tested negative and were quarantined until their symptoms resolved.

At the girls camp, no campers or staff became symptomatic during the first session. One female camper became symptomatic while travelling by car to camp after receiving a negative at-home PCR test. She was administered a rapid antigen test from her local pharmacy which came back positive, and she went home prior to arrival. Four weeks later she attended the second session after resolution of her COVID-19 infection.

During the beginning of the second session, one unvaccinated camper (Camper A) developed a fever two days after arriving at camp. She was immediately quarantined and administered a COVID-19 rapid antigen test which resulted negative. She continued to be isolated from other campers and staff. Two of her cabin mates who were also unvaccinated (Campers B and C) had shared a tent with her and were quarantined separately from Camper A. Campers B and C did not receive a COVID-19 rapid antigen test because they were asymptomatic. Other campers within Camper A's cohort who did not share a tent with her were quarantined separately as a group because they were vaccinated and remained asymptomatic for 5 days following exposure (Campers D, E, and F).

24 hours after developing symptoms, Camper A was administered another COVID-19 rapid antigen test which came back positive this time. At this point, the parents of Campers A, B, and C were contacted. All three campers were picked up by their parents before the in-camp testing timepoint. After leaving camp, 5 days after exposure, both unvaccinated campers developed symptoms of COVID-19. Camper B tested positive while Camper C declined testing. No other campers became symptomatic in the week following exposure and no other rapid tests were administered at this time. Fig 4 is a visual representation of the quarantine and testing procedures conducted for this positive COVID-19 case.

Throughout the rest of the second session, 5 other campers and 2 staff members from the girls camp were administered a rapid test for symptoms; all results came back negative. A summary of symptomatic testing results can be found in Table 5. Throughout the entire summer, at both the boys and girls camps, no campers or staff became symptomatic during a wilderness trip. there was one staff member that self-reported a positive case after the camp session had

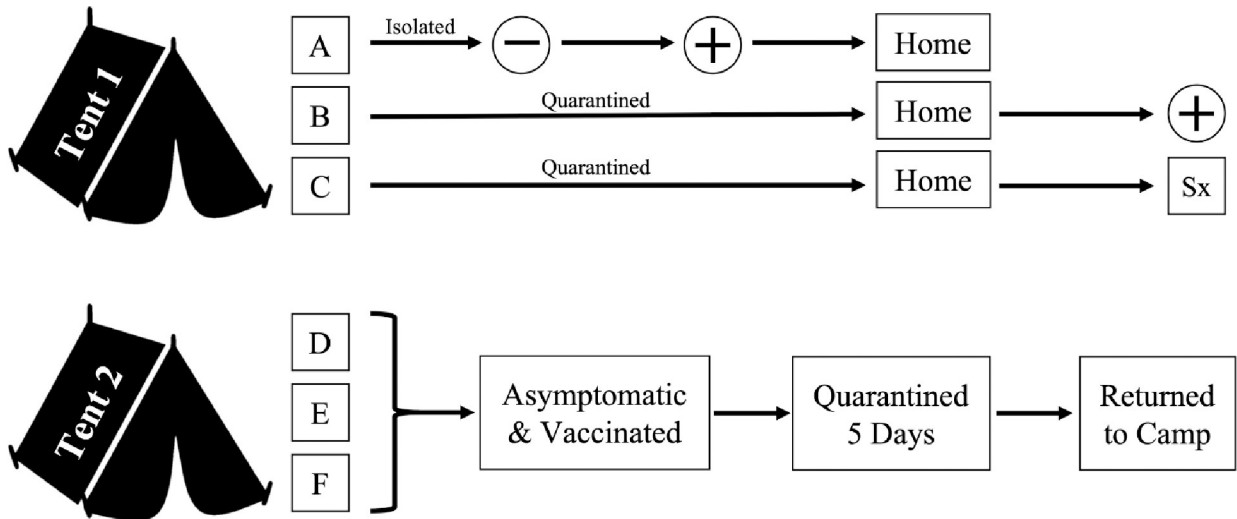

**Fig 4. Quarantine and testing procedures initiated after a positive COVID-19 case at camp.**

ended. Based on the timeframe of symptoms, it was not clear whether this case was camp-related, and the result was not included in analysis.

## Discussion

Mitigation of COVID-19 spread was successful at two overnight summer camps following proper testing and cohorting procedures as outlined in 2021 guidance from state and national policy recommendations. The goals of the prevention policy included mitigating risks of exposure and limiting spread in the event of a positive case. These policies allowed the camp to act as a closed 'bubble' within which COVID-19 precautions could be lifted for the duration of the season. Once lifted, the protocols were robust and there were no additional COVID-19 infections detected at camp or on wilderness trips. The foundation of this policy included hiring individuals only if they were fully vaccinated against the virus. Among the staff, there were no confirmed cases or breakthrough cases that infiltrated the camp. This is owed in part to guidelines for leaving campgrounds which included mandatory masks and hand hygiene when leaving the camp 'bubble'. Coupled with proactive symptom monitoring, there were no positive COVID-19 cases detected in staff.

This study adds to an expanding literature demonstrating the efficacy of non-pharmacologic interventions in preventing the spread of COVID-19 and guiding policy for potential

**Table 5. Results of symptomatic testing for COVID-19 during camp sessions.**

| | Boys Camp | |
|---|---|---|
| | First Session | Second Session |
| | Pos/Neg | Pos/Neg |
| Staff | 0/2 | 0/0 |
| Campers | 0/0 | 0/0 |
| | Girls Camp | |
| | First Session | Second Session |
| | Pos/Neg | Pos/Neg |
| Staff | 0/0 | 0/2 |
| Campers | 0/0 | 1/5 |

future outbreaks. According to a survey by the American Camping Association, interventions such as cohorting, frequent handwashing, and mask wearing were all common during the 2020 summer season [15, 16]. However, this is the first study that utilized a rigorous COVID-19 testing strategy to create a "bubble", which allowed campers to then safely forego cohorting and mask wearing.

The findings of this report are limited by a lack of COVID-19 testing at the end of the camp term. It is generally understood that COVID-19 infection in this age group have less severe symptoms and can even be asymptomatic. Considering the one case of a staff member developing symptoms after camp ended, there may have been asymptomatic transmission of the COVID-19 virus within the camp. While national COVID-19 numbers had a decline in late spring and early summer 2021, by the time camp was concluding these numbers were again increasing due to the spread of the Delta variant. To our knowledge, no other person became symptomatic after returning home from camp premises. However, exit testing could have captured asymptomatic circulating infection.

Another limitation to this study is the criteria used for a probable COVID-19 case and when a camper is tested. At the wilderness camps, there was a confirmed non-COVID upper respiratory infection (URI) that spread surreptitiously. These individuals would report to the infirmary and symptoms would not match the common COVID-19 profile, and therefore were assumed to have the non-COVID URI. An exit test could have ruled out non-standard presentations of COVID-19 infection prevalent among the campers. Unknown policy adherence is another limitation, as there was no direct measure of adherence to camp prevention guidelines. The transmission of the non-COVID URI is assumed to have the same contagion risks and routes as COVID-19.

The findings of this study support a robust protocol that could be expanded to large groups and reinforce the validity of the closed "bubble" system in the prevention of infectious spread [17]. To properly propagate this bubble, pre-arrival testing and quarantining should be required. Proper masking, hand hygiene, disinfection protocols, and social distancing are also pillars of an infection mitigation plan. Cohorting at arrival can potentially limit spread in the event of a positive case. Post-arrival testing ensures no newly acquired virus has infiltrated the bubble. so that infection prevention precautions can be relaxed. With mindful execution, these summer wilderness camps were able to prevent spread and allow for a successful camp season in 2021. It is the hope of the authors that this article will assist camps in creating a successful prevention strategy for COVID-19 and other future outbreaks.

## Acknowledgments

The authors wish to acknowledge the work of the camp directors and camp staff for their hard work that made the summer of 2021 possible.

## Author Contributions

**Conceptualization:** Tirzah Weiss, Tate Reuter, Evan Dowell, Mitchell Singstock, Katherine Smith, Jeffrey Schlaudecker.

**Data curation:** Tirzah Weiss, Tate Reuter, Evan Dowell, Mitchell Singstock, Katherine Smith, Jeffrey Schlaudecker.

**Formal analysis:** Tirzah Weiss, Tate Reuter, Evan Dowell, Mitchell Singstock, Katherine Smith, Jeffrey Schlaudecker.

**Funding acquisition:** Jeffrey Schlaudecker.

**Investigation:** Tirzah Weiss, Tate Reuter, Evan Dowell, Mitchell Singstock, Katherine Smith, Jeffrey Schlaudecker.

**Methodology:** Tirzah Weiss, Tate Reuter, Evan Dowell, Mitchell Singstock, Katherine Smith, Jeffrey Schlaudecker.

**Project administration:** Evan Dowell, Katherine Smith, Jeffrey Schlaudecker.

**Resources:** Jeffrey Schlaudecker.

**Supervision:** Jeffrey Schlaudecker.

**Validation:** Jeffrey Schlaudecker.

**Writing – original draft:** Tirzah Weiss, Tate Reuter, Evan Dowell, Mitchell Singstock, Katherine Smith, Jeffrey Schlaudecker.

**Writing – review & editing:** Tirzah Weiss, Tate Reuter, Evan Dowell, Mitchell Singstock, Katherine Smith, Jeffrey Schlaudecker.

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
