## [Decision Letter · Decision Letter 0]

21 Jun 2023

PONE-D-23-04916Evaluation of an Infection Control Protocol to Limit COVID-19 at Residential Summer Camps in 2021PLOS ONE

Dear Dr. Singstock,

Thank you for submitting your manuscript to PLOS ONE. After careful consideration, we feel that it has merit but does not fully meet PLOS ONE’s publication criteria as it currently stands. Therefore, we invite you to submit a revised version of the manuscript that addresses the points raised during the review process.

We look forward to receiving your revised manuscript.

Kind regards,

Olushayo Oluseun Olu

Academic Editor

PLOS ONE

Reviewers' comments:

Reviewer's Responses to Questions

**Comments to the Author**

1. Is the manuscript technically sound, and do the data support the conclusions?

Reviewer #1: Yes

Reviewer #2: Partly

2. Has the statistical analysis been performed appropriately and rigorously? 

Reviewer #1: Yes

Reviewer #2: No

3. Have the authors made all data underlying the findings in their manuscript fully available?

Reviewer #1: Yes

Reviewer #2: Yes

4. Is the manuscript presented in an intelligible fashion and written in standard English?

Reviewer #1: Yes

Reviewer #2: Yes

5. Review Comments to the Author

Reviewer #1: Introduction

Well-written and flow of ideas is appropriate.

1. I recommend adding a paragraph about vaccine hesitancy (VH) among this age group, which made it even harder to accept vaccination especially at the beginning of rollout (May 10th, 2021). Indicate the relationship of VH and herd immunity.

Kindly add this as a reference: Musa S, Dergaa I, Abdulmalik MA, Ammar A, Chamari K, Saad HB. BNT162b2 COVID-19 Vaccine Hesitancy among Parents of 4023 Young Adolescents (12-15 Years) in Qatar. Vaccines (Basel). 2021 Sep 2;9(9):981. doi: 10.3390/vaccines9090981. PMID: 34579218; PMCID: PMC8473301.

2. Add another paragraph about the vulnerability to COVID-19 infection among different immune status, chronic disease, mental health, age, socioeconomic, nationality, gender, evidence from previous literature. [you need to link the interpretation of your study results with these factors, later in the discussion part].

Methods

Under “Vaccination policy”

1. Can you specify what do you mean by fully vaccinated? Two doses, three doses, type of validated vaccines, duration from vaccination (6 month/9 months/12 months).

2. How did you ensure that home test is carried out correctly to avoid measurement bias? Was there any training, or short video provided for participants?

3. Using the word ‘administered’ for the covid-19 test gives ambiguous meaning. You may replace throughout the manuscript with either “tested” or “undertaken”.

Under “Camper characteristics”

1. Kindly clarify the travel policy for those campers coming from Midwest region and outside US during the time of the airplane travel, in regard requirements of vaccination, quarantine and 24-48 hours negative PCR or rapid test requirements.

2. You have collected data on vaccination status, what about history of previous covid-19 infection and recovery? It is also an important information especially within first 6 months in terms of neutralizing antibodies and protection.

Under “COVID-19 vaccination status”

Kindly provide interpretation of the higher rates of vaccination among older age group. Previous literature in relation to vaccine hesitancy will assist including below citation:

Musa S, Dergaa I, Abdulmalik MA, Ammar A, Chamari K, Saad HB. BNT162b2 COVID-19 Vaccine Hesitancy among Parents of 4023 Young Adolescents (12-15 Years) in Qatar. Vaccines (Basel). 2021 Sep 2;9(9):981. doi: 10.3390/vaccines9090981. PMID: 34579218; PMCID: PMC8473301.

Under “discussion”

1. Will you explain how the staff were trained and prepared for receiving/handling mitigative procedures.

2. In addition, will you mention the successful adoption of the buddle system in the prevention and control of COVID-19 within other mass gathering for example among athletes/ world cup and compare the effectiveness of various interventions.

These citations will help you:

https://www.researchgate.net/publication/358949547_Olympic_Games_in_COVID-19_times_lessons_learned_with_special_focus_on_the_upcoming_FIFA_World_Cup_Qatar_2022

https://www.researchgate.net/publication/358949547_Olympic_Games_in_COVID-19_times_lessons_learned_with_special_focus_on_the_upcoming_FIFA_World_Cup_Qatar_2022

3. How the results of this study will aid in the planning within other settings, cohorts, and future emergencies. (Add a section about im

Reviewer #2: General comments:

The study had great potential to contribute to the evidence base to support the impact of safety precautions or protocols for infection control, especially for COVID-19 where there have been controversies, mixed with adherence issues. The meticulous attempt to implement and evaluate a well-crafted protocol is highly commendable.

However, the paper is weak in providing strong statistical argument to support their findings. For a start, a measure of adherence to the protocol itself is an important piece in substantiating the results of the protocol’s effectiveness in preventing spread of COVID-19. Though this was pointed out by the authors themselves, along with other limitations that were mentioned including not conducting repeat tests at the end of the camping.

Although not stated, this study was an interventional study of a quasi-experimental design, where there was no control group, but there was a form of pre-/post-intervention assessment of the outcome variable, COVID-19 test positivity.

Comments on specific sections/sub-sections are as follows:

Methods - The components of the Protocol (that is the ‘intervention’) that was implemented were excellently described, however could be intentional in indicating them as subsumed under this intervention. The authors should consider this revision to display a sub-section under the Methods section. This could substitute for the table on the list of CDC-recommended procedures for camps that was displayed in the introduction section, which could simply be referenced.

Pre-Camp COVID-19 Testing Policies - Authors could rephrase the statement that ‘parents were required to administer an at-home test via saliva collection’ as in essence it was an at-home sample collection that was done for a PCR test.

COVID-19 Vaccination Status - Clarity should be provided on the numbers in parenthesis in Line 2.

Symptomatic Testing for COVID-19 - The word “denied” in the sentence “Camper B tested positive while Camper C denied testing” should be clarified.

Discussion - It is not conventional to report additional even if pertinent findings in the Discussion section of original research papers. For instance, the authors reported the ‘one person on staff that self-reported a positive case after the camp ended’. Though it made for good reading and complimented the narrative on the study limitations, the Editor may wish to provide more guidance in keeping with the journal’s policy or style. Perhaps what was more glaring was the general absence of inferences on how the findings compare with other reports of evaluation of infection control protocols for COVID-19 among similar study populations. Authors should consider this revision to the discussion section.

6. PLOS authors have the option to publish the peer review history of their article (what does this mean?). If published, this will include your full peer review and any attached files.

Reviewer #1: **Yes: **Dr. Sarah Musa

Reviewer #2: **Yes: **Seye Babatunde

---

## [Author Response · Author response to Decision Letter 0]

6 Aug 2023

Dear Editors, 

Thank you for reviewing our manuscript for publication. Please find our response to the reviewer’s comments below. 

Reviewer #1 

Introduction 

I recommend adding a paragraph about vaccine hesitancy (VH) among this age group, which made it even harder to accept vaccination especially at the beginning of rollout (May 10th, 2021). Indicate the relationship of VH and herd immunity. 

Kindly add this as a reference: Musa S, Dergaa I, Abdulmalik MA, Ammar A, Chamari K, Saad HB. BNT162b2 COVID-19 Vaccine Hesitancy among Parents of 4023 Young Adolescents (12-15 Years) in Qatar. Vaccines (Basel). 2021 Sep 2;9(9):981. doi: 10.3390/vaccines9090981. PMID: 34579218; PMCID: PMC8473301. 

Thank you for your recommendation. We added a paragraph discussing vaccination rates and hesitancy to our introduction and included the provided references. Please see line 103. 

Add another paragraph about the vulnerability to COVID-19 infection among different immune status, chronic disease, mental health, age, socioeconomic, nationality, gender, evidence from previous literature. [you need to link the interpretation of your study results with these factors, later in the discussion part]. 

The authors felt that this was outside the scope of this paper. Our goal was to describe an infectious disease control protocol and the authors feel that an in-depth description of the epidemiology of COVID-19 vulnerabilities would not substantively contribute to the objectives of this paper. 

Methods 

Can you specify what do you mean by fully vaccinated? Two doses, three doses, type of validated vaccines, duration from vaccination (6 month/9 months/12 months). 

Our protocol specified fully vaccinated as two doses of either the Pfizer-BioNTech or Moderna vaccine. We have clarified this in line 130 under “Vaccination Policy.” 

How did you ensure that home test is carried out correctly to avoid measurement bias? Was there any training, or short video provided for participants? 

Training for saliva collection was provided with the test package instructions. We have clarified this in line 138 under “Quarantine and Pre-Camp COVID-19 Testing Policies.” 

Using the word ‘administered’ for the covid-19 test gives ambiguous meaning. You may replace throughout the manuscript with either “tested” or “undertaken.” 

We prefer to use the word administered because campers completed the rapid test under the guidance of a camp medic. 

Kindly clarify the travel policy for those campers coming from Midwest region and outside US during the time of the airplane travel, in regard requirements of vaccination, quarantine and 24-48 hours negative PCR or rapid test requirements. 

All campers and their families were requested to wear a face mask and avoid public gatherings during the two weeks before arriving at camp, and a KN95 mask was provided for the camper to wear while traveling to and from camp, as described in “Quarantine and Pre-Camp COVID-19 Testing Policies.” Testing and quarantine policies did not vary based on the camper’s method of travel to camp. 

You have collected data on vaccination status, what about history of previous covid-19 infection and recovery? It is also an important information especially within first 6 months in terms of neutralizing antibodies and protection. 

We agree that prior infection conveys protection. However, we did not collect this information from campers. 

Kindly provide interpretation of the higher rates of vaccination among older age group. Previous literature in relation to vaccine hesitancy will assist including below citation: Musa S, Dergaa I, Abdulmalik MA, Ammar A, Chamari K, Saad HB. BNT162b2 COVID-19 Vaccine Hesitancy among Parents of 4023 Young Adolescents (12-15 Years) in Qatar. Vaccines (Basel). 2021 Sep 2;9(9):981. doi: 10.3390/vaccines9090981. PMID: 34579218; PMCID: PMC8473301. 

Thank you for this recommendation. We have included a discussion on vaccine hesitancy in the introduction (see line 105). A primary driver in this study was the FDA approval for different age groups. 

Discussion 

Will you explain how the staff were trained and prepared for receiving/handling mitigative procedures. 

Infirmary staff of the camps handled mitigating procedures. A description of their training is beyond the scope of this paper. 

In addition, will you mention the successful adoption of the buddle system in the prevention and control of COVID-19 within other mass gathering for example among athletes/ world cup and compare the effectiveness of various interventions. These citations will help you: https://www.researchgate.net/publication/358949547_Olympic_Games_in_COVID-19_times_lessons_learned_with_special_focus_on_the_upcoming_FIFA_World_Cup_Qatar_2022

We believe our research adds to the literature of the bubble system in mitigating disease spread. We have included a brief discussion on this topic to the discussion. A comparison of the effectiveness of various interventions was beyond the scope of this report. 

How the results of this study will aid in the planning within other settings, cohorts, and future emergencies. 

Thank you for this suggestion. Please see the rewritten and expanded final paragraph. 

Reviewer #2 

General 

The study had great potential to contribute to the evidence base to support the impact of safety precautions or protocols for infection control, especially for COVID-19 where there have been controversies, mixed with adherence issues. The meticulous attempt to implement and evaluate a well-crafted protocol is highly commendable. 

However, the paper is weak in providing strong statistical argument to support their findings. For a start, a measure of adherence to the protocol itself is an important piece in substantiating the results of the protocol’s effectiveness in preventing spread of COVID-19. Though this was pointed out by the authors themselves, along with other limitations that were mentioned including not conducting repeat tests at the end of the camping. 

Although not stated, this study was an interventional study of a quasi-experimental design, where there was no control group, but there was a form of pre-/post-intervention assessment of the outcome variable, COVID-19 test positivity. 

We appreciate these general comments. We agree that this research does constitute an experiment without control and, therefore, lacks rigorous statistical comparison. We provide reference to other camps, where applicable, to describe the incidence of COVID-19 without protective measures. Additionally, it is a limitation that we did not retest as campers left; however, we did closely monitor for COVID-19 symptoms. Therefore, we feel confident that if a camper or staff member was symptomatic for COVID-19, we would have identified them. 

Methods 

The components of the Protocol (that is the ‘intervention’) that was implemented were excellently described, however could be intentional in indicating them as subsumed under this intervention. The authors should consider this revision to display a sub-section under the Methods section. This could substitute for the table on the list of CDC-recommended procedures for camps that was displayed in the introduction section, which could simply be referenced. 

We appreciate the reviewer's comments. Given the frequently updated recommendations and changing knowledge since the beginning of the pandemic, we felt that a tabular representation of recommended procedures added to the generalizability and would enable readers to understand the state of science more easily in May 2021. 

Pre-Camp COVID-19 Testing Policies - Authors could rephrase the statement that ‘parents were required to administer an at-home test via saliva collection’ as in essence it was an at-home sample collection that was done for a PCR test. 

This has been rephrased. Please see line 141. 

COVID-19 Vaccination Status - Clarity should be provided on the numbers in parenthesis in Line 2. 

This clarification has been made. Please see line 212. 

Symptomatic Testing for COVID-19 - The word “denied” in the sentence “Camper B tested positive while Camper C denied testing” should be clarified. 

This wording has been modified. 

Discussion 

It is not conventional to report additional even if pertinent findings in the Discussion section of original research papers. For instance, the authors reported the ‘one person on staff that self-reported a positive case after the camp ended’. Though it made for good reading and complimented the narrative on the study limitations, the Editor may wish to provide more guidance in keeping with the journal’s policy or style. Perhaps what was more glaring was the general absence of inferences on how the findings compare with other reports of evaluation of infection control protocols for COVID-19 among similar study populations. Authors should consider this revision to the discussion section. 

We have made this edit to the discussion. Thank you for your comments.

---

## [Decision Letter · Decision Letter 1]

21 Sep 2023

PONE-D-23-04916R1Evaluation of an Infection Control Protocol to Limit COVID-19 at Residential Summer Camps in 2021PLOS ONE

Dear Dr. Singstock,

Thank you for submitting your revised manuscript to PLOS ONE. After careful consideration, we feel that you have not comprehensively addressed the comments of the two reviewers as it currently stands. Therefore, we invite you to submit a revised version of the manuscript that addresses the points raised during the second review process.

We look forward to receiving your revised manuscript.

Kind regards,

Olushayo Oluseun Olu

Academic Editor

PLOS ONE

Reviewers' comments:

Reviewer's Responses to Questions

**Comments to the Author**

1. If the authors have adequately addressed your comments raised in a previous round of review and you feel that this manuscript is now acceptable for publication, you may indicate that here to bypass the “Comments to the Author” section, enter your conflict of interest statement in the “Confidential to Editor” section, and submit your "Accept" recommendation.

Reviewer #1: (No Response)

Reviewer #2: All comments have been addressed

2. Is the manuscript technically sound, and do the data support the conclusions?

Reviewer #1: Partly

Reviewer #2: Yes

3. Has the statistical analysis been performed appropriately and rigorously? 

Reviewer #1: Yes

Reviewer #2: Yes

4. Have the authors made all data underlying the findings in their manuscript fully available?

Reviewer #1: No

Reviewer #2: Yes

5. Is the manuscript presented in an intelligible fashion and written in standard English?

Reviewer #1: No

Reviewer #2: Yes

6. Review Comments to the Author

Reviewer #1: Thank you for highlighting the raised concerns, kindly for some points, the author response was insufficient to meet the review requirements.

For instance, comment #2

Vulnerability due to coexisting conditions is an important parameter related to susceptibility and risk, hence the rate of infection which varies greatly between healthy and non-healthy individuals and so the prevention and control protocol.

Under methods, comment #2

The review point asked to explain the travel policy, PCR testing, vaccination requirement and quarantine related to participants who travelled to the location of the camp. However, the authors have explained about the face mask! Strict requirements were introduced all over the world during the time of your study, even some countries have banned travels, hence, it is still important to clarify the raised points.

Also for comment #1 discussion:

The authors response indicate that training of the main study intervention is not an important source of bias that could threaten the internal validity of the study. This is not out of scope concern, however, it is amongst the core requirement. Kindly describe how training of staff who applied the prevention strategies was carried out including the duration and numbers. How have you ensured that application of these strategies was on uniform and in place. (monitoring of intervention is an important indicator of outcome measure).

The review aims to improve the quality, representation and coherence of the paper, therefore, it is important to response adequately to every raised concern. Thanks

Reviewer #2: The paper presented the implementation of a COVID-19 prevention protocol that attempted a “closed ‘bubble’ system” and was evaluated for its effectiveness by monitoring pre- and post-intervention COVID-19 PCR test and symptomatic test positivity. Key study limitations in the study design and statistical analysis were also clearly identified and discussed. The authors have done well to address the concerns that were pointed out by the reviewers and resubmitted a revised manuscript.

However, a mistaken notion regarding ‘adherence’ to the protocol has not been addressed. On one hand, the authors rightly pointed out that “there was no measure of adherence to camp prevention guidelines [the protocol]”, which is an important variable in substantiating the results of the protocol’s effectiveness in preventing spread of COVID-19. The import of this is that adherence cannot be ‘reported’ as a result or an outcome, even if it was guaranteed during the implementation of the guidelines. However, in the Abstract, ‘adherence’ to the protocol was reported in the “results” as successful; it was also emphatically stated in the “conclusion” that adherence led to the study outcome, even though the level of adherence was not measured.

It is suggested that, without the benefit of a measure of adherence but instead a description of the steps taken, the statements regarding adherence should be rephrased in the paper, particularly in the abstract. For instance, the statements in lines 56, 61 & 63 that begin with “adherence to [the protocol]…” could be re-written as “implementation of [the protocol]...”

7. PLOS authors have the option to publish the peer review history of their article (what does this mean?). If published, this will include your full peer review and any attached files.

Reviewer #1: **Yes: **Sarah Rashid Musa

Reviewer #2: **Yes: **Seye Babatunde

---

## [Author Response · Author response to Decision Letter 1]

16 Oct 2023

Dear Editors,

Thank you for reviewing our manuscript for publication. Please find our response to the reviewer’s comments below.

Have the authors made all data underlying the findings in their manuscript fully available?

 The data is freely available through dryad with a link included in the data availability statement. 

Reviewer #1: 

Thank you for highlighting the raised concerns, kindly for some points, the author response was insufficient to meet the review requirements.

For instance, comment #2

Vulnerability due to coexisting conditions is an important parameter related to susceptibility and risk, hence the rate of infection which varies greatly between healthy and non-healthy individuals and so the prevention and control protocol.

 Thank you for your feedback. We have added the following statement: “While healthy children are generally less likely to suffer severe symptoms of COVID-19 when compared to those with pre-existing medical conditions and the elderly, they are still vulnerable to potentially serious complications of illness and can spread the coronavirus throughout their communities. In children in the United States, the baseline rate of hospitalizations from COVID-19 was 8 in 100,000; however, these rates increase in children with obesity, chronic lung disease, and/or who are black [8].”

Under methods, comment #2

The review point asked to explain the travel policy, PCR testing, vaccination requirement and quarantine related to participants who travelled to the location of the camp. However, the authors have explained about the face mask! Strict requirements were introduced all over the world during the time of your study, even some countries have banned travels, hence, it is still important to clarify the raised points.

 Thank you for your request for clarification. Our paper currently explains how campers arrived in a variety of transportation methods and any travel outside of the camp was only permitted for staff, which is described in the methods. We explained how PCR testing was performed at home and at camp for campers who were not fully vaccinated with 2 doses of either Pfizer or Moderna vaccines. We additionally describe our quarantine policy and cohorting. 

Also for comment #1 discussion:

The authors response indicate that training of the main study intervention is not an important source of bias that could threaten the internal validity of the study. This is not out of scope concern, however, it is amongst the core requirement. Kindly describe how training of staff who applied the prevention strategies was carried out including the duration and numbers. How have you ensured that application of these strategies was on uniform and in place. (monitoring of intervention is an important indicator of outcome measure).

We have added this to the section on In-Camp Testing “All saliva collection was overseen by the medical staff or by staff members who were directly trained by the medical staff on how to collect saliva for PCR testing.” We also added that " Staff members were taught the in-camping cohorting guidelines prior to each camp session. Reminders were provided during each meal. Additionally, staff members were encouraged to correct campers and staff members who were not following masking or cohorting guidelines.”

We have eliminated the use of the word adherence from the article, because this was not an outcome directly measured. We appreciate your feedback. 

The review aims to improve the quality, representation and coherence of the paper, therefore, it is important to response adequately to every raised concern. Thanks

Reviewer #2: 

The paper presented the implementation of a COVID-19 prevention protocol that attempted a “closed ‘bubble’ system” and was evaluated for its effectiveness by monitoring pre- and post-intervention COVID-19 PCR test and symptomatic test positivity. Key study limitations in the study design and statistical analysis were also clearly identified and discussed. The authors have done well to address the concerns that were pointed out by the reviewers and resubmitted a revised manuscript.

However, a mistaken notion regarding ‘adherence’ to the protocol has not been addressed. On one hand, the authors rightly pointed out that “there was no measure of adherence to camp prevention guidelines [the protocol]”, which is an important variable in substantiating the results of the protocol’s effectiveness in preventing spread of COVID-19. The import of this is that adherence cannot be ‘reported’ as a result or an outcome, even if it was guaranteed during the implementation of the guidelines. However, in the Abstract, ‘adherence’ to the protocol was reported in the “results” as successful; it was also emphatically stated in the “conclusion” that adherence led to the study outcome, even though the level of adherence was not measured.

It is suggested that, without the benefit of a measure of adherence but instead a description of the steps taken, the statements regarding adherence should be rephrased in the paper, particularly in the abstract. For instance, the statements in lines 56, 61 & 63 that begin with “adherence to [the protocol]…” could be re-written as “implementation of [the protocol]...”

 Thank you for your feedback. We agree that we cannot make statements about the efficacy of adherence to the protocol when adherence was not a measured outcome but was subsumed under the umbrella of COVID-19 cases. We have eliminated the word “adherence” from the article.

---

## [Editor Report · Decision Letter 2]

31 Oct 2023

PONE-D-23-04916R2Evaluation of an Infection Control Protocol to Limit COVID-19 at Residential Summer Camps in 2021PLOS ONE

Dear Dr. Singstock,

Thank you for submitting your manuscript to PLOS ONE. After careful consideration, we feel that it has merit but does not fully meet PLOS ONE’s publication criteria as it currently stands. Therefore, we invite you to submit a revised version of the manuscript that addresses the points raised during the review process.

Many thanks for finally addressing the comments of the reviewers which I believe has improved the quality of your manuscript. I have three minor suggestions before final consideration of the document for acceptance as follows:

1. Please include very clear goal and objectives of your study at the end of the introduction section

2. Move the sub-sections "Camper's Characteristics" and "Covid-19 Vaccination Status" from the methods to the beginning of the results section.

We look forward to receiving your revised manuscript.

Kind regards,

Olushayo Oluseun Olu

Academic Editor

PLOS ONE

Journal Requirements:

Additional Editor Comments:

Many thanks for finally addressing the comments of the reviewers which I believe has improved the quality of your manuscript. I have three minor suggestions before final consideration of the document for acceptance as follows:

1. Please include very clear goal and objectives of your study at the end of the introduction section

2. Move the sub-sections "Camper's Characteristics" and "Covid-19 Vaccination Status" from the methods to the beginning of the results section.

Thank you

---

## [Author Response · Author response to Decision Letter 2]

2 Nov 2023

Thank you for your recommendations. I have added the following statement to the end of the introduction section, “The purpose of this study was to assess the effectiveness of an infection control protocol developed to mitigate the spread of COVID-19 at two multi-week residential summer camps in 2021. The primary outcome of interest, therefore, is COVID-19 transmission rates within the summer camp.” Additionally, the camper characteristic and COVID-19 vaccination status sections have been moved to results.

---

## [Editor Report · Decision Letter 3]

7 Nov 2023

Evaluation of an Infection Control Protocol to Limit COVID-19 at Residential Summer Camps in 2021

PONE-D-23-04916R3

Dear Dr. Singstock,

We’re pleased to inform you that your manuscript has been judged scientifically suitable for publication and will be formally accepted for publication once it meets all outstanding technical requirements.

Kind regards,

Olushayo Oluseun Olu

Academic Editor

PLOS ONE
---

## [Editor Report · Acceptance letter]

14 Nov 2023

PONE-D-23-04916R3 

Evaluation of an Infection Control Protocol to Limit COVID-19 at Residential Summer Camps in 2021 

Dear Dr. Singstock:

I'm pleased to inform you that your manuscript has been deemed suitable for publication in PLOS ONE. Congratulations! Your manuscript is now with our production department. 

Kind regards, 

on behalf of

Dr. Olushayo Oluseun Olu 

Academic Editor

PLOS ONE